LORM: a novel reinforcement learning framework for biped gait control

Zhang Weiyi
Jiang Yancao
http://orcid.org/0000-0002-0178-8130 Farrukh Fasih Ud Din
http://orcid.org/0000-0001-9791-4500 Zhang Chun zhangchun@tsinghua.edu.cn
Zhang Debing
Wang Guangqi
School of Integrated Circuits, Tsinghua University , Beijing , People’s Republic of China
Zhang Qichun
Electronic publication date: 2022 Mar 28
Publication date: 2022
Volume: 8
Electronic Location ID: e927
Received 2021 Oct 4; Accepted 2022 Feb 28
Copyright: © 2022 Zhang et al.
Copyright year: 2022
Copyright holder: Zhang et al.
License: This is an open access article distributed under the terms of the Creative Commons Attribution License, which permits unrestricted use, distribution, reproduction and adaptation in any medium and for any purpose provided that it is properly attributed. For attribution, the original author(s), title, publication source (PeerJ Computer Science) and either DOI or URL of the article must be cited.
License URL: https://creativecommons.org/licenses/by/4.0/

Keywords: Gait controlling, Robotic, Reinforcement learning

Funding: National Natural Science Foundation of China U20A20220 This work was supported by the National Natural Science Foundation of China (No. U20A20220). The funders had no role in study design, data collection and analysis, decision to publish, or preparation of the manuscript.

==============================
Legged robots are better able to adapt to different terrains compared with wheeled robots. However, traditional motion controllers suffer from extremely complex dynamics properties. Reinforcement learning (RL) helps to overcome the complications of dynamics design and calculation. In addition, the high autonomy of the RL controller results in a more robust response to complex environments and terrains compared with traditional controllers. However, RL algorithms are limited by the problems of convergence and training efficiency due to the complexity of the task. Learn and outperform the reference motion (LORM), an RL based framework for gait controlling of biped robot is proposed leveraging the prior knowledge of reference motion. The proposed trained agent outperformed the reference motion and existing motion-based methods. The RL environment was finely crafted for optimal performance, including the pruning of state space and action space, reward shaping, and design of episode criterion. Several improvements were implemented to further improve the training efficiency and performance including: random state initialization (RSI), the noise of joint angles, and a novel improvement based on symmetrization of gait. To validate the proposed method, the Darwin-op robot was set as the target platform and two different tasks were designed: (I) Walking as fast as possible and (II) Tracking specific velocity. In task (I), the proposed method resulted in the walking velocity of 0.488 m/s, with a 5.8 times improvement compared with the original traditional reference controller. The directional accuracy improved by 87.3%. The velocity performance achieved 2× compared with the rated max velocity and more than 8× compared with other recent works. To our knowledge, our work achieved the best velocity performance on the platform Darwin-op. In task (II), the proposed method achieved a tracking accuracy of over 95%. Different environments are introduced including plains, slopes, uneven terrains, and walking with external force, where the robot was expected to maintain walking stability with ideal speed and little direction deviation, to validate the performance and robustness of the proposed method.

Introduction

Autonomous robots have been used to perform different tasks and help to reduce workloads. Robotic arms are widely used in factories, significantly improving productivity. For those non-moving robotics, motion planning is to control all the joints of the robot to achieve the target position. Currently, various motion controlling and optimization methods have been proposed and widely used (Sucan, Moll & Kavraki, 2012; Huda et al., 2020; Ratliff et al., 2009; Liu & Liu, 2021). Another family of robots, mobile robots, are widely used to perform tasks including serving, rescue, and medical treatment. For mobile robots, the controlling task includes both motion and moving, which increases the complexity of the design. Wheeled robots have the simplest moving pattern. However, wheeled robots are not capable of complex outdoor environments where the ground may slope or be uneven. Biped robots have more lifelike movements and can walk on more complex terrains, showing great potential in nursing, rescue, and other various applications. However, the gait planning of biped robots is more difficult due to the extremely complex dynamics properties. Although different methods have been proposed to control biped gait, including model-based methods (Chevallereau et al., 2013), bio-inspired methods (Liu et al., 2020), and machine-learning-based methods (Wang, Chaovalitwongse & Babuska, 2012), gait controlling of biped robots remains one major task for robotics.

Traditional gait control methods are generally based on dynamics and mathematical models. The inverted pendulum is one of the most adapted models which has inspired many algorithms such as those of Pratt et al. (2006), Lee & Goswami (2007), Li, Li & Cui, (2016), and Kajita et al. (2010). Furthermore, the Zero Momentum Point (ZMP) (Vukobratovic et al., 2012) was proposed as a way to control different humanoid robots. In addition to walking on the plain, some traditional methods were proposed to solve the gait planning on slopes, uneven terrains, or with the disturbance of external force (Kim, Park & Oh, 2007; Morisawa et al., 2012; Yi et al., 2016; Yi, Zhang & Lee, 2010; Smaldone et al., 2019). Morisawa et al. (2012) proposed an improved balance control algorithm based on the Capture Point (CP) controller, enabling the robot to walk on uneven terrain. Yi et al. (2016) proposed an algorithm enabling the robot to walk on unknown uneven terrain while minimizing the modification to the original gait on plain based on online terrain estimation. Based on similar models for balance, Gong et al. (2019) proposed an algorithm enabling the robot Cassie to ride a segway. However, there exist some disadvantages of traditional methods. Firstly, traditional methods highly rely on dynamics and mathematical models for both the robot and the terrain, requiring large amounts of time and labor for designers. When the type of the robot or the property of the terrain is changed, the model needs to be re-designed. This disadvantage also results in the lack of adaptivity of the gait controller. Furthermore, the performance of the human-designed model is also limited by the prior knowledge and experience of the designers, which cannot fully explore the potential of the robot.

Reinforcement learning (RL) is suitable for controlling tasks where the agent can be trained within the designed environment as a controller for different tasks (Arulkumaran et al., 2017; Gullapalli, Franklin & Benbrahim, 1994; Johannink et al., 2019). In those tasks, the trained model can instruct the controlled servos and adjust to the change of environment simultaneously, which results in better adaptivity and autonomy. In addition, many works have proved that exploration during the training process enables the controller to obtain skills beyond human knowledge, resulting in much better performance (Silver et al., 2017). With the development of efficient, tiny neural networks and the hardware accelerators, reinforcement learning can easily be implemented on robotic devices with embedded processors (Zhang et al., 2021; Meng, Kuppannagari & Prasanna, 2020). While the motion controlling tasks can be solved by reinforcement learning with high performance, gait controlling of legged robots remains a challenge due to the complexity. Different algorithms are used to solve the biped tasks from OpenAI Gym (Brockman et al., 2016) by implementing pure reinforcement learning to control the joints directly (Heess et al., 2017). However, when these algorithms are used in the gait controlling for a real robot with much more complex dynamics property, the convergence and performance suffer. In addition, the trained model may also have a nonhuman-like gait, which is usually unacceptable. Thus, training frameworks leveraging prior knowledge have become an important point of study for RL-based gait controllers. One solution is to combine reinforcement learning with the traditional models (Lin et al., 2016; Phaniteja et al., 2017; Jiang et al., 2020). Gil, Calvo & Sossa (2019) proposed a multi-level algorithm to train the RL-based controller, by finding stable poses that meet the condition of ZMP. Then a motion sequence composed of the poses is trained to form a gait cycle. Liu, Ning & Chen (2018) and Jiang et al. (2020) used reinforcement learning to optimize the parameters of the dynamics models improving the performance. However, the used traditional models, such as static walking patterns, largely limit the performance in velocity. And in many works, the joints are not directly controlled by the reinforcement learning algorithms, thus the flexibility and adaptivity of reinforcement learning decrease. Some knowledge including the symmetry of locomotion can also improve the performance of the trained model (Yu, Turk & Liu, 2018). Other works utilize existing gait as the prior knowledge. Xi & Chen (2020) proposed a hybrid reinforcement learning framework to keep the center of pressure (CoP) close to the reference gait, achieving an adaptive gait on both static and dynamic platforms. Imitation-learning-based algorithms such as GAIL (Ho & Ermon, 2016) are also widely used to improve the training efficiency and overcome the problem of nonhuman-like motion. Imitation learning can mimic the motion of humans or robots controlled by traditional algorithms in much fewer training iterations with considerable performance and convergence (Peng et al., 2018; Peng et al., 2017; Xie et al., 2018; Xie et al., 2019). However, imitation learning seeks to master the given motion, and the performance of the trained model is similar to expert data, with the potential of robot unexplored. Xie et al. (2018) proposed an imitation-learning-based algorithm for the biped control with different velocities. However, The reference action is artificially modified for different velocities, and the feasibility cannot be guaranteed. Thus the robot failed to achieve the highest velocity. Meanwhile, the agent is trained only on the plain, with the adaptivity and robustness of reinforcement learning not fully explored. We proposed a novel framework for biped control known as Learn and Outperform the Reference Motion (LORM) to fully explore the robot’s potential and utilize the prior data from the reference motion.

The main contributions of this paper include: LORM, an RL-based controlling framework, was proposed for biped gait. Using both simulation environment state and reference state, the RL agent can not only mimic the reference motion but also explore a much better policy by combining the environment reward and imitation reward. LORM uses one gait cycle of the reference motion without modification for different tasks and terrains, which can be collected by various methods such as motion capture. With the assistance of the reference motion, LORM has much better training efficiency and performance than the pure RL method. Instead of simply imitating, in the tasks of walking as fast as possible and tracking specific velocity, LORM significantly outperformed the reference motion with better performance in velocity, direction, and robustness on different terrains. By the simple reward shaping, LORM can obtain different velocities from the same reference motion. With the full exploration by reinforcement learning, our max velocity on the plain is more than 2× compared with the rated highest speed of the official document of the robot, whose gait is generated by traditional methods. Compared with other proposed traditional or reinforcement learning control algorithms, the velocity performance is even more outstanding, with more than 7× improvement. For other complex terrains, the proposed algorithm also has novel performance, with approximately 4× improvement on slopes and uneven terrains. To our knowledge, LORM achieves the highest velocity on the platform robot Darwin-op (Ha, Tamura & Asama, 2013). In addition to the high max velocity, the velocity can be adjusted at the training stage flexibly, providing more choices for different tasks and environments.

An RL environment with finely crafted state space, action space, reward, and criterion was proposed to allow the RL agent to learn how to control the robot in an expected manner with high performance.

Several tricks were introduced into the framework. Random state initialization (RSI) and Gaussian noise were introduced into the training process. We proposed an improvement fully leveraging the symmetry of the gait cycle to further improve the performance and training efficiency.

Sufficient validation environments were constructed in Webots (Michel, 2004) to evaluate the performance and robustness of the proposed method and can be used to evaluate other methods and robots.

Methods

Reinforcement learning

Overall theory

Reinforcement learning aims to generate an agent to provide proper actions according to the observations from the environment. A reinforcement learning structure is composed of an agent to be trained and the environment for training and evaluating. As shown in Fig. 1, the agent repeatedly interacts with the environment in RL training, according to the value function generated by a neural network (Mnih et al., 2013; Mnih et al., 2015) or a simple Q-value table that stores the expected reward for each combination of the state and action. In trajectory T (from the beginning state to the end state of the environment), at the t-th time step, the agent receives observation ot from the environment, which is based on current state st of environment and outputs action at accordingly. The environment receives the action and gets into the next state st+1 and outputs reward rt for the action and new observation ot+1. After some iterations of interaction, the agent will update its weights to get a higher total reward Rtotal=∑t=1nrt, where n stands for the number of steps in one trajectory. This may be achieved by either getting higher reward in every step or by trying to maintain more steps before the end of the trajectory.

Figure 1 Basic reinforcement learning framework.

Policy gradient

Policy gradient algorithms are one mainstream family of reinforcement learning algorithms that are capable of complex continuous tasks. Policy gradient algorithms generate optional actions directly without one function or table indicating the reward for different actions. The output of the network is the probability distribution of actions.

(1) p(a|s,θ)=P[At=a|St=s,θt=θ]

where θ is the current weights of the neural network. The agent will choose the action according to the probability distribution at every step. Thus the probability P(τ|θ) of trajectory τ = (s1, a1,…, sT, aT) and the total reward can be calculated as:

(2) P(τ|θ)=p(s1)∏t=1Tpθ(at|st,θ)p(st+1|st,at)

(3) R(τ)=∑t=1Trt

Given N different trajectories τ1,τ2,…,τN sampled by policy π with weights θ, the expectation of the total reward for one trajectory can be approximated as:

(4) R¯θ=∑τR(τ)P(τ|θ)≈1N∑n=1NR(τn)

The training will optimize the weights θ of policy π to obtain the highest expected reward.

(5) θ∗=arg⁡maxθ⁡R¯θ

During the gradient ascent of backward propagation, the probabilities of actions bringing more rewards will increase while other probabilities are reduced. The gradient can be calculated as:

(6) ∇R¯θ≈1N∑n=1NR(τn)∇log⁡P(τn|θ)=1N∑n=1NR(τn)∑t=1Tn∇log⁡p(atn|stn,θ)=1N∑n=1N∑t=1TnR(τn)∇log⁡p(atn|stn,θ)

Advantage function Aθ(st, at) is introduced replacing R(τ) to further improve the training efficiency:

(7) ∇R¯θ≈1N∑n=1N∑t=1TnAθ(st,at)∇log⁡p(atn|stn,θ)

(8) Aθ(st,at)=Gt−b

(9) Gt=∑t′=tTnγt′−trt′n

where Gt is the discounted accumulated reward, γ is the discounted coefficient, and b is the regularizer which will be further discussed in the next subsection.

Actor-critic structure

Actor-critic structure introduces one critic that calculates the value function Vθ(s) of the state s indicating whether the state is good or bad. The value function is used in the calculation of the advantage function. By the prediction of critic, the advantage function of each state can be calculated respectively without the information of the whole trajectory, as shown in Eq. (10). Here rt + Vθ(st+1) acts as the discounted accumulated reward Gt and Vθ(st) acts as regularizer b.

(10) Aθ(st,at)=rt+Vθ(st+1)−Vθ(st)

The critic is a neural network in deep reinforcement learning, which will be trained together with the agent (called actor) during the training process. The loss for critic can be calculated by temporal-difference (TD) approach:

(11) Loss=Vθ−γVθ(st+1)−rt

Proximal policy gradient

The sampling is time-consuming in the training of reinforcement learning. However, the sampled trajectories expired after the update of policy network π and new trajectories need to be sampled based on the new policy, which resulted in low trajectory utilization and training efficiency. Thus importance sampling is introduced in TRPO (Schulman et al., 2015a) and PPO (Schulman et al., 2017) to use the trajectories repeatedly, improving the training efficiency.

Importance sampling is widely used to estimate the expectation where the sampling of the distribution is difficult to obtain. Given one distribution x∼p(x), the expectation f(x) can be estimated as Eq. (12), where xi is sampled from p(x).

(12) Ex∼p[f(x)]=1n∑if(xi)

However, when the sample of xi is difficult to access, importance sampling can be used by introducing another distribution x∼q(x) with a relatively small difference with p(x). The expectation f(x) with distribution p(x) can be calculated as Eq. (13), where xi is sampled from q(x). Based on importance sampling, PPO uses one neural network to interact with the environment collecting samplings and another neural network to update weights, which is called an off-policy algorithm. The training efficiency and sample utilization are largely improved. In addition, the difference of distribution p(x) and q(x) should be small to ensure the accuracy.

(13) Ex∼p[f(x)]=Ex∼q[f(x)p(x)q(x)]=1n∑if(xi)p(xi)q(xi)

PPO maintains two actors πθ(Pi) and πθ′ (OldPi) with the same network structure. The actor πθ is used to update weights while the πθ′ is used to interact with the environment sampling the training data. The gradient for optimization is calculated based on importance sampling:

(14) ∇R¯θ=Eτ∼pθ(τ)[R(τ)∇log⁡pθ(τ)]=Eτ∼pθ′(τ)[pθ(τ)pθ′(τ)R(τ)∇log⁡pθ(τ)]

PPO maintains one critic neural network for state evaluation. Introducing advantage function to replace R(τ), the gradient is further calculated as:

(15) ∇R¯θ=E(st,at)∼πθ[Aθ(st,at)∇log⁡pθ(atn|stn)]=E(st,at)∼πθ′[Pθ(st,at)Pθ′(st,at)Aθ(st,at)∇log⁡pθ(atn|stn)]≈E(st,at)∼πθ′[Pθ(st,at)Pθ′(st,at)Aθ′(st,at)∇log⁡pθ(atn|stn)]=E(st,at)∼πθ′[pθ(at|st)pθ′(at|st)pθ(st)pθ′(st)Aθ′(st,at)∇log⁡pθ(atn|stn)]≈E(st,at)∼πθ′[pθ(at|st)pθ′(at|st)Aθ′(st,at)∇log⁡pθ(atn|stn)]

The objective function for gradient ascent can be derived from ∇Rθ:

(16) Jθ′(θ)=E(st,at)∼πθ′[pθ(at|st)pθ′(at|st)Aθ′(st,at)]

The clip coefficient ε is furthermore introduced to limit the update step, which can keep the difference between Pi and OldPi small enough for importance sampling. The final objective function for gradient ascent is calculated as Eq. (17), where k denotes the number of current iteration. The overall structure of PPO is shown in Fig. 2. In this work, PPO from OpenAI Baselines (Dhariwal et al., 2017) is implemented as the training algorithm with hyperparameters as shown in Table 1. The neural network structure of Pi and OldPi is shown in Fig. 3, and is composed of one input layer, one output layer, and two hidden layers. The algorithm also introduced generalized advantage estimation (GAE) (Schulman et al., 2015b) in the calculation of advantage function, improving the training performance.

(17) JPPOθk(θ)=∑(st,at)min(pθ(at|st)pθk(at|st)Aθk(st,at),clip(pθ(at|st)pθk(at|st),1−ε,1+ε)Aθk(st,at))

Table 1 Hyperparameters of PPO.

Name	Description	Value	
timesteps_per_actorbatch	Time steps between two updates of policy neural network	2,048	
clip_param	The clipping parameter ε	0.1	
optim_epochs	The number of epochs in one optimization cycle	10	
optim_stepsize	The stepsize of the optimizer	0.0001	
optim_batchsize	The batchsize of the optimizer	64	
gamma	Discount coefficient	0.99	
lambda	Coefficient of generalized advantage estimation	0.95	

Figure 2 Framework of PPO.

Figure 3 Neural network structure of Pi and OldPi.

General framework of LORM

LORM is an improved feedback control framework based on reinforcement learning, with full use of the prior knowledge of the reference motion. One cycle of the reference motion is generated by methods such as motion capture, from which information of state and action at each time step can be accessed before the training of the RL agent. The workflow of LORM is shown in Fig. 4. In each time step, the reference motion provides both recorded state s^ and action a^ to the RL agent. The input of the RL agent is the combination of reference state s^ and the state s received from the simulation environment. In addition, a one-dimension extra state indicating the phase of gait cycle is introduced into state space, which is called the assistant state. The RL agent generates the corresponding bias δa as the output. The target action a is calculated as the sum of reference action a^ and bias δa generated by RL agent. A PD-controller is used to control the servo motors according to target action a. Different from Xie et al. (2018) in which the robot learns the velocity specified by the reference motion, no change to the reference motion is introduced for all different velocities and environments in our work. We argue that by changing the reward function slightly, the robot can learn different velocity in various environments, and explore the highest potential velocity automatically without any change of reference motion. In our work, the reference motion is not the final goal but only an example of the gait from which the robot learns a basic pattern of gait and surpasses it largely in velocity, accuracy of direction, and adaptability to various environments after training. To achieve the proposed result, the RL environment is finely crafted and is discussed in the following subsections.

Figure 4 General framework of LORM.

State space and action space

The input state of the RL agent is divided into three parts: environment state received from the simulation environment, reference state recorded from the reference motion and assistant state showing the current phase of the gait cycle. The 35-D environment state consists of diverse state information such as the current position, the orientation of the robot base and the angles of joints, as shown in Table 2. The reference state is recorded in the same format, thus has the same 35 dimensions as the environment state. The assistant state is one integer ranging from 0 to 17, indicating the current phase of the gait cycle, which has 18 phases in total. Thus the gait cycle is a finite state machine (FSM) with 18 states. The total dimension of state space is 35 + 35 + 1 = 71.

Table 2 Environment state.

Name	Dimension	Description	
Base Position	3	The xyz position of the base of robot	
Orientation	3	The Euler angle of the robot	
Velocity	3	The current xyz velocity	
Angular velocity	3	The current xyz angular velocity	
Position of center of mass	3	The xyz position of the center of mass	
Angles of joints	20	The current angles of the joints, indicating the current posture	
Total	35	–	

To reduce the dimension of action space a, the set of controlled joints is pruned according to the influence of each joint for walking task in one reference gait cycle. The changes of joint angles along with the phase of gait cycle is shown in Fig. 5. The joints such as ArmUpperR, ArmUpperL, ArmLowerR, and ArmLowerL keep unchanged throughout the cycle. This indicates they have little contribution to the walking task and can be removed from the controlled set. The set of controlled joints after pruning is shown in Table 3, which is denoted by action vector δa.

Figure 5 Joint angles of reference motion in one cycle.

Table 3 Joints after pruning.

Name	Name	Description	
ShoulderR	ShoulderL	Joints on shoulders with one DoF (Degree of Freedom)	
		(Different from human, one another DoF is eliminated)	
PelvR	PelvL	Joints on hips with two DoF	
LegUpperR	LegUpperL	(Different from human, one another DoF is eliminated)	
LegLowerR	LegLowerL	Joints on knees with one DoF	
		(The same as human)	
AnkleR	AnkleL	Joints on feet with two DoF	
RootR	RootL	(The same as human)	

To fully leverage the reference motion, our method generates the target action based on both the output of agent δa and corresponding reference motion a^. Different from earlier works generating target action by RL agent directly, the output action bias δa of RL agent is not the final target angles of the joints. In this case, the goal of the RL agent is to make a suitable adjustment to the reference action according to the current state. The final 12-D target action a explicitly indicating the target angles of joints is calculated as the sum of reference action and output bias of the RL agent:

(18) a=a^+δa

Reward function

Inspired by Peng et al. (2018), the reward function of LORM is divided as task reward and imitation reward:

(19a) R=wIRI+wTRT

(19b) wI+wT=1

RT denotes the task reward, indicating the performance of the walking trajectory, while RI denotes the imitation reward, indicating the similarity between the state s of the simulation environment and the reference state s^ at the same phase. wT and wI are the weights balancing the influence of each reward, which add up to 1.

Task reward

Task reward, including velocity reward and deviation penalty, was used to prompt the agent to interact properly with the environment and complete the task. The velocity reward encouraged the robot to walk at the expected velocity stably. Two different velocity rewards were designed according to the tasks. For the task of walking as fast as possible, the velocity reward was calculated as shown in Eq. (20).

(20) rsim=50×min(xpos−x^pos,Lmax)

where xpos denotes the position in the forward direction at the current time-step while x^pos denotes the position at the last time-step. Lmax is introduced as the ceiling of the reward to guarantee stability. The velocity reward for the task of tracking specific velocity is calculated as shown in Eq. (21a):

(21a) rsim=exp[−105×(xpos−x^pos−xt)2]

(21b) xt=vt×t

where xpos and x^pos have the same meaning as Eq. (20). xt is the expected forward distance in one time step, which can be converted from target velocity vt by Eq. (21b), where t denotes the length of one time step.

The deviation penalty is implemented to guarantee the accuracy of direction. When the robot deviates from the target direction, the total reward will decrease. The deviation penalty is defined as follows:

(22) Lossy={0,−20≤pitch≤20,−1,else.

The deviation is not significant when the pitch of the robot is in the range from −20 to 20, thus the penalty is zero. When the absolute value exceeds this range, the penalty is set to −1 to command the robot to adjust the direction. The total environment reward is the sum of velocity reward and deviation penalty:

(23) RT=rsim+Lossy

Imitation reward

Imitation reward RI encourages the robot to imitate the reference motion in each phase of the gait cycle. In this case, the policy is constrained by the prior information obtained from the reference, which significantly improves the training efficiency. Imitation is especially important at the beginning of the training process where little knowledge is accumulated and the reward is almost zero in every training epoch. The imitation reward is composed of imitation rewards for joints, the center of mass, and Euler angles:

(24) RI=wItrIt+wIcrIc+wIorIo

where wtI, wcI and woI are separate weights of the three parts of the reward. Based on sufficient experiments, weights are set as follows: wtI = 0.6, wcI = 0.3 , woI = 0.1.

The reward for joints is calculated as shown in Eq. (25), where q^ti and qit respectively stand for the angles of the i-th joint of the reference and the simulation environment, wit is the weight indicating the importance of the i-th joint. Different joints contribute differently to the walk. For instance, the joints of ankles are more important than joints of shoulders. Thus the weights of left and right shoulder are set to 0.05 while other joints of legs are 0.09, and add up to 1.

(25) rIt=exp[−∑iwit(||q^ti−qti||2)]

The reward for the center of mass is calculated as shown in Eq. (26), where q^cj and qjc denotes the reference and simulation position of the center of mass at the current time-step.

(26) rIc=exp[−∑j(||q^cj−qcj||2)]

The reward for the Euler angle is calculated as shown in Eq. (27), where q^oj and qjo denotes the reference and simulation Euler angle at the current time-step.

(27) rIo=exp[−∑k(||q^ok−qok||2)]

Criterion for done

The robot interacts with its environment during each training period collecting the states, actions and rewards with which to update network weights. When the episode ends, signal Done is set as 1 to reset the world and begin a new episode. If the criterion for Done is unsuitable, the robot will collect information of poor conditions, such as falling down, deviating, or marching in unnatural manners like crawling. These unexpected data will mislead the RL agent during the training, especially at the beginning of the training process. A strict criterion for signal Done is introduced to maintain the purity of the experience pool, which will end the episode in advance before the bad experience is collected. The criterion is shown in Eq. (28). If the current time-step t exceeds the max time-step number Tlimited, the episode will end automatically. Other stop conditions include: (a) The height of center of mass hcom is lower than the threshold Hlimit. (b) The roll angle exceeds the threshold. (c) The yaw angle exceeds the threshold. These three conditions indicate the robot has lost balance and the experience is misleading to the training.

(28) Criterion={Truet>TlimitTruehcom<HlimitTrueroll>45°orroll<−45°Trueyaw>45°oryaw<−45°Falseelse

Methods of improvement

Three main methods were introduced into the framework to improve the training efficiency and the performance of the trained model: (a) Random state initialization. (b) Symmetrization of actions and states. (c) Noise in training.

Random state initialization

The environment state is initialized at the beginning of every episode. Traditional RL training is based on fixed state initialization (FSI), resetting the environment to a fixed beginning state. In such methods, agent will learn the policy serially. For example, in the walking task, the robot will learn to stand stable at the beginning, then learn the next sub-task of walking. However, at the beginning of the training, the robot falls directly in most episodes, thus the latter sub-tasks cannot be trained, resulting in a low training efficiency. To make full use of the reference motion and improve the training efficiency, the random state initialization (RSI) is introduced into training. This method has been widely adopted (Peng et al., 2018; Nair et al., 2018). At the beginning of one episode, the environment is initialized as one random phase of the reference gait cycle. In this way, the agent learns earlier and later sub-tasks simultaneously with a much higher efficiency.

Symmetrization of actions and states

The pattern of gait is defined as the cycling switch of the swing leg and instance leg between the two legs. In the first half of the cycle, the left leg is the swing leg and in the second half of the cycle the right leg is the swing leg. The two halves are symmetrical, which provides useful prior knowledge for the training. The symmetrization of reference and the symmetrization of state and action are proposed to improve the training efficiency and performance.

Symmetrization of reference: The gait cycle of reference motion is not symmetric, decreasing the performance of the trained agent, as shown in Fig. 5, where the curves of AnkleR and AnkleL are not perfectly symmetrical. To achieve absolute symmetry, the phases 9–17 are derived based on the phases 0–8, leveraging the symmetry of the two halves of the gait cycle. The resulting gait cycle has a perfect mirror symmetry in corresponding phase pairs: 0–9, 1–10, and so on.

Symmetrization of state and action: The state space is composed of the phase information (1D), the simulation state (35D) and the reference state (35D). The robot only needs to learn the first half of the gait cycle due to the symmetry of the gait cycle, then the second half can be generated according to symmetry. We propose the compression of state space and action space to accelerate the training process based on this innovation as shown in Fig. 6. At phases 0–8, the states are unchanged and are sent to the RL agent directly, while states at phases 9–17 are processed into the symmetry state as follows: For the simulation state, mirror symmetry to the 20-dimension joint angles was obtained replacing the original ones. Other dimensions remained unchanged. Then the processed symmetry state was sent to the RL agent. The 20-dimension joint angles of the reference state is also replaced by the symmetric ones. Thus, all states indicating joint angles are symmetric. For states at symmetric phase pairs, such as phases 0 and 9, 1 and 10, the output action generated by the RL agent will be highly similar. At phase 0–8, the output of the RL agent will be sent to the adder directly. At phase 9–17, the output of the RL agent will be recovered by symmetrization firstly. Then the symmetry output will be sent into the adder. The final target angles for the joints are the sum of the output action and the reference angles for both the first and second halves of gait cycle.

Figure 6 Flowchart of state symmetrization.

Noise in training

Gaussian noise was introduced into the output of the RL agent in the training process. During training, the RL agent was trained to overcome uncertain noise. Thus the training efficiency slightly decreased while the robustness in various environments and resistance to the external force of the trained model were significantly improved.

Results

Parameter selection

In the LORM framework, different parameters were tested and selected to achieve the best performance. Weights of imitation/task reward wI, wT and max forward length in one time step Lmax are discussed in this subsection.

The comparison of different imitation/task rewards is shown in Fig. 7, where the task is walking as fast as possible: (a) The average reward of one episode. (b) The velocity of trained models. When the imitation reward weight was low (wI = 0.60, wT = 0.40), the robot was instructed to ignore the reference motion and concentrate on interacting with the environment. Thus the training efficiency decreased due to the low utilization of the prior knowledge from the reference motion. However, when the environment reward weight was low (wI = 0.65, wT = 0.35), the robot learned more from the reference motion, resulting in a low exploration beyond the reference motion, and the performance suffered. The combination of wI = 0.62, wT = 0.38 achieved the best trade-off of training efficiency and performance.

Figure 7 Comparison of different weights wI, wT.

The comparison of different max forward length Lmax is shown in Fig. 8, where the task was walking as fast as possible: (a) The average length of one episode. (b) The average reward of one episode. In the training, Lmax greater than 0.02 resulted in the loss of balance because the robot concentrated on the velocity and ignored the need to maintain balance. When Lmax is small, the velocity is limited, resulting in low reward with the potential of robot incompletely exploited. Lmax = 0.02 achieves the best performance, as shown in previous experiments.

Figure 8 (A–B) Comparison of different max step length Lmax.

Random initialization

The comparison of training with RSI and FSI is shown in Fig. 9: (a) Average length in one interaction episode. (b) Average reward in one interaction episode. RSI clearly outperformed FSI in training efficiency and performance.

Figure 9 (A–B) Comparison of length and reward in one episode between FSI and RSI.

Symmetrization

Symmetrization is implemented in the reference motion and state & action for two different tasks: (a) Walking as fast as possible. (b) Tracking specific velocity.

The speed comparison of different symmetrization methods is shown in Fig. 10: Symmetrization was firstly implemented in the reference motion, which achieved better velocity performance. Then symmetrization to state & action was implemented, further improving the performance. The average speeds of the different methods are shown in Table 4. The method implementing symmetrization in both reference motion and state & action achieved the best speed of 0.488 m/s, with a 578% improvement compared with the original reference motion and a 34% improvement compared with the method without symmetrization.

Figure 10 Speed comparison of different symmetrization methods.

Table 4 Speed of different methods.

	Average forward distance in one time step (m)	Average speed (m/s)	
Original reference motion	0.0023	0.072	
LORM without symmetrization	0.0117	0.365	
LORM with symmetrization in reference motion	0.0131	0.411	
LORM with symmetrization in reference motion and state & action	0.0156	0.488	

The results of the different methods for tracking specific speeds are shown in Fig. 11. The tracked forward distance in one time step is shown as xt = 0.01 m in (a) and xt = 0.005 m in (b), which can also be represented as 0.313 m/s and 0.156 m/s. In (a), the method with symmetrization kept the speed of 0.301 m/s with an accuracy of 96.3%, which has 5.6% improvement over the method without symmetrization. In (b), the method with symmetrization kept the speed of 0.163 m/s with an accuracy of 95.8%, which has 6.8% improvement than method without symmetrization.

Figure 11 (A–B) Comparison of tracking accuracy between methods with and without symmetrization.

The comparison of training efficiency between methods with and without symmetrization is shown in Fig. 12, where the task is walking as fast as possible in (a) and tracking specific speed in (b). Training efficiency improved significantly in both tasks due to the reduction of environment complexity and the prior knowledge of symmetrization.

Figure 12 (A–B) Comparison of training efficiency.

Noise in training

Gaussian noise was introduced during training into the 12 joints important for walking task, with mean μ = 0 and standard deviation σ = 0.01. Two different environments were designed for the evaluation: (a) The same Gaussian noise was introduced into the joints. (b) The same Gaussian noise and random external force were introduced. The deviations in different training and evaluating environment are shown in Table 5. The max deviation was 0.17 m in the evaluating environment (a) and 0.31 in the evaluating environment (b) when noise was introduced in the training, which respectively decreased by 61.4% and 55.7% compared with the model trained without noise.

Table 5 Direction deviations of different methods.

	Evaluating environment (a)	Evaluating environment (b)	
Training without noise	0.44 m	0.70 m	
Training with noise	0.17 m	0.31 m	

Performance and robustness in different environments

In this subsection, all improvements were introduced into the method to achieve the final result. To validate the performance and robustness of the proposed method, different walking environments were constructed including plain, slope, uneven terrain, and external force.

Walking on plains

The position and orientation during the walking procedure are shown in Fig. 13, where the task was to walk as fast as possible. The robot achieved an average speed of 0.488 m/s, with 22.48 m forward distance in 1,440 time steps. The velocity performance achieved a 5.8 time improvement compared with the reference motion generated by the traditional controller. The absolute deviation was 0.31 m and the relative deviation was 1.4%. The relative deviation decreased 87.3% compared with the original reference motion, which is 11.0%. In addition, the max Euler angle deviation from the expected direction (euler_y in Fig. 13B) was 16.2°, however, the robot corrected the direction automatically and simultaneously to maintain accuracy while the reference motion kept the deviated direction with no adjustment. The comparison of velocity performance is shown in Table 6. The compared five works include: the official document of Darwin-op, Li, Li & Cui, 2016, Gil, Calvo & Sossa (2019), and two different algorithms in Xi & Chen (2020), in which the target platform is the same as or similar with our platform. Our work and the first two compared works are tested on the platform Darwin-op, while the last three works on the platform NAO. For Xi & Chen (2020), the highest velocity among various cases is selected as the absolute velocity on the plain. Normalization of the velocity is introduced to ensure the precision of the comparison, which divides the absolute velocity by the height of the controlled robot. The normalized velocity of our work is 2 times compared with the rated max velocity, and approximately 10 times compared with other algorithms. The result shows that our work fully explored the potential of the robot, getting better performance compared with traditional algorithms and other reinforcement learning algorithms combined with dynamics models. To our knowledge, we have achieved the highest velocity on the platform Darwin-op.

Figure 13 (A–B) Position and orientation during walking procedure.

Table 6 Comparison of velocity performance on plain.

	Ours	Rated max velocity of Darwin-op	Li’s	Gil’s	Xi’s	Model-based in Xi’s	
Absolute velocity (m/s)	0.488	0.24	0.06	0.0615	0.069	0.061	
Height of robot (m)	0.454	0.454	0.454	0.58	0.58	0.58	
Normalized velocity (s−1)	1.075	0.529	0.132	0.106	0.119	0.105	

For the task of tracking specific speed, the comparison of the actual speed and target speed is shown in Fig. 14, where the equivalent target speed (forward distance in one time step) is xt = 0.01 m in (a) and xt = 0.005 m in (b). The tracking accuracy achieved above 95% in both conditions.

Figure 14 (A–B) Comparison of real velocity and tracked velocity.

Walking on slopes

The sequence of walking uphill is shown in Fig. 15, where the slope was 0.1 rad. The task was tracking specific velocity, rather than walking as fast as possible, to ensure the stability of walking. The robot kept the velocity of 0.295 m/s, with an accuracy of 94.0% compared with the target velocity of 0.313 m/s. The directional error was adjusted simultaneously, with a max value of 25.4°. The sequence of walking downhill is shown in Fig. 16, where the slope was 0.1 rad and the task was tracking specific velocity. The robot maintained a velocity of 0.278 m/s, with an accuracy of 88.8% compared with the target velocity of 0.313 m/s. The slight drop in accuracy was caused by the high velocity when walking downhill, which results in greater difficulty to keep balance. The velocity performance outperforms the rated max velocity on plain though the velocity is limited to around 0.313 m/s during the training. The comparison of velocity performance on slopes is shown in Table 7, where the compared works come from Xi & Chen (2020). The velocity performance is more than 5 times compared with the reference work. The robot maintained the directional error within 18° in the tested 1,440 time steps, which is better than the original reference motion. In addition, the robot adjusted its pose before losing balance to maintain a stable walking gait, which is the benefit of RL based controller.

Figure 15 The sequence of walking uphill.

Figure 16 The sequence of walking downhill.

Table 7 Comparison of velocity performance on slopes.

		Ours	Xi’s	Model-based in Xi’s	
Uphill	Absolute velocity (m/s)	0.295	0.037	0.028	
	Height of robot (m)	0.454	0.58	0.58	
	Normalized velocity (s− 1)	0.612	0.119	0.105	
Downhill	Absolute velocity (m/s)	0.278	0.069	0.061	
	Height of robot (m)	0.454	0.58	0.58	
	Normalized velocity (s− 1)	0.612	0.119	0.105	

Uneven terrain

To further evaluate the robustness of the proposed method, an environment with the uneven ground was constructed in Webots as shown in Fig. 17. The walking sequence is shown in Fig. 18. For the task of walking as fast as possible, the robot achieved a max velocity of 0.453 m/s, with a max position deviation of 0.234 m, and a max direction deviation of 17.5°. The comparison of velocity performance is shown in Table 8. The three compared works come from Liu, Ning & Chen, 2018, Yi et al. (2016), and Morisawa et al. (2012). The velocity performance is largely improved compared with the reference works. For the task of tracking specific velocity, the robot maintained the velocity of 0.299 m/s, with an accuracy of 95.5% compared with the target velocity 0.313 m/s. The max position deviation was 0.385 m and the max directional deviation was 21.4°. The trained robot was able to adapt well to the uneven ground and complete both tasks with high quality.

Figure 17 Uneven terrain.

Figure 18 Sequence of walking on uneven terrain.

Table 8 Comparison of velocity performance on uneven terrain.

	Ours	Liu’s	Yi’s	Morisawa’s	
Absolute velocity (m/s)	0.453	0.022	0.05	0.267	
Height of robot (m)	0.454	0.58	1.61	1.54	
Normalized velocity (s−1)	0.998	0.038	0.031	0.173	

Series of slopes

A series of slopes combining plain-uphill-plain-downhill-plain was designed to further validate the adaptability of the proposed method. The walking sequence is shown in Fig. 19. The robot automatically adjusted to the gait policy to adapt different slopes and perform better. For instance, the angles of AnkleR and AnkleL during walking are shown in Fig. 20. The angle of AnkleR is higher during walking uphill and lower downhill. Correspondingly, the angle of AnkleL is lower during walking uphill and higher during the downhill. Both two ankles lift higher when the robot is walking uphill and lift lower when downhill.

Figure 19 Sequence of walking on series of slopes.

Figure 20 Angle of two ankles during the walking sequence.

External force

In the validation of resistance to external force, an environment providing external force for eight time steps in every 200 time steps was constructed. The robot was able to walk stably with the external force up to 10N from forward and back, or up to 6N from the left of right. The Euler angle of the robot is shown in Fig. 21 and the linear velocity is shown in Fig. 22. The robot made adjustments to recover the sudden change of the Euler angle and linear velocity caused by a random external force. The robot achieved a velocity of 0.363 m/s, which is also improved significantly from the original reference motion. The comparison of performance with other works is shown in Table 9. The compared works come from Liu, Ning & Chen (2018) and Smaldone et al. (2019). Our work has a resistance to stronger external force from both x-axis and y-axis with a better velocity performance.

Figure 21 Orientation curve of walking with external force.

Figure 22 Linear velocity curve of walking with external force.

Table 9 Comparison of performance with external force.

	Ours	Liu’s	Smaldone’s	
Max force in x-axis (N)	[−10, 10]	0	[−3, 6]	
Max force in y-axis (N)	[−6, 6]	∼6.5	[−1, 6]	
Absolute velocity (m/s)	0.363	0.032	0.05	
Height of robot (m)	0.454	0.58	0.58	
Normalized velocity (s−1)	0.80	0.055	0.086	

Variety of capable environments

Our method is capable in the widest range of environments including: plains, slopes, uneven terrains and walking with random disturbance of external force. The comparison of capable environments is shown in Table 10, where the compared works are those mentioned in the previous comparisons.

Table 10 Comparison of capable environments.

	Ours	Li’s	Gil’s	Xi’s	Liu’s	Yi’s	Morisawa’s	Smaldone’s	
Plains	✓	✓	✓	✓	✓	✓	✓	✓	
Slopes	✓			✓					
Uneven terrains	✓				✓	✓	✓		
External force	✓				✓			✓	

Conclusions

A novel framework for biped gait controlling was proposed based on reinforcement learning. The advantages of LORM include: 1. Better performance in velocity and direction. Our work achieved the best velocity performance on the platform Darwin-op to our knowledge. In addition, the robot can adjust to the environments to maintain balance and a stable velocity and direction simultaneously. 2. Compared with pure RL-based methods, the training efficiency and convergence are largely improved by leveraging the supervision of reference motion. 3. Compared with imitation learning based methods, LORM significantly outperforms expert data instead of simply imitating. 4. LORM is capable of different environments including: plain, slopes, uneven terrains, and walking with the disturbance of external force. A training environment was expertly crafted for the RL agent to learn the expected gait manner proficiently. In addition, various methods were introduced to simplify the task and make full use of the prior knowledge of reference motion, improving the performance of the proposed method further. To validate the proposed method, different tasks and environments were designed, which can also be used in the validation of other methods or robots in the future. The experiment was conducted on the Darwin-op platform, which adapts servos as joints instead of a hydraulic actuator. Thus, the gait controller of different small or medium-sized robots can also be trained with the proposed method with adjustment of training parameters. Thus, the design of the gait controller is much more efficient, costing less time and manpower. The proposed framework has been implemented on gait controlling. Additional tasks and actions such as rolling, jumping will be researched with the proposed framework in future studies. By fusing different actions into one single RL controller, the robot will act more naturally with an easy and fluent change between different motions.

Supplemental Information

Supplemental Information 1 The files and codes for training and evaluation.

This archive includes the .wbt files to be opened by Webots. By opening the file “/worlds/darwin2_motion.wbt”, the walking process of robot can be shown. By opening the file “/worlds/darwin2_motion_training.wbt”, the training process can be started.

The codes are also included.

More details are described in Readme.txt .

Click here for additional data file.

Supplemental Information 2 All raw data and codes for processing the raw data.

The raw data and the python codes to plot the figures that appear in the article. For all provided results such as deviation, we also provide the original record of the walking process of the robot and the processing code.

Click here for additional data file.

We would like to thank the anonymous reviewers for their valuable suggestions.

Additional Information and Declarations

Competing Interests

Author Contributions

Data Availability

The authors declare that they have no competing interests.

Weiyi Zhang conceived and designed the experiments, performed the experiments, analyzed the data, performed the computation work, prepared figures and/or tables, authored or reviewed drafts of the paper, and approved the final draft.

Yancao Jiang conceived and designed the experiments, performed the experiments, analyzed the data, performed the computation work, prepared figures and/or tables, authored or reviewed drafts of the paper, and approved the final draft.

Fasih ud Din Farrukh performed the computation work, prepared figures and/or tables, authored or reviewed drafts of the paper, and approved the final draft.

Chun Zhang performed the experiments, analyzed the data, performed the computation work, authored or reviewed drafts of the paper, and approved the final draft.

Debing Zhang performed the experiments, prepared figures and/or tables, and approved the final draft.

Guangqi Wang performed the experiments, prepared figures and/or tables, and approved the final draft.

The following information was supplied regarding data availability:

The codes and environments for training and evaluation and the raw data, with codes for analysis and plotting figures, are available in the Supplemental Files.

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
