# Peer review of "LORM: a novel reinforcement learning framework for biped gait control"

_PeerJ Computer Science, doi:10.7717/peerj-cs.927_

## Round 0.1 · original submission · Major Revisions

The manuscript is well-written with an interesting topic. However, you need to compare your study to existing methods to highlight the contribution of the proposed algorithm. Additionally, the typos should be corrected via careful proof reading before re-submission.

Reviewer 1 ·

Basic reporting

This paper proposes a biped control framework based on reinforcement learning. The expert trajectory of the traditional controller is introduced to accelerate the training. And the exploration of reinforcement learning ensures the final model outperforms the expert instead of simply imitating the expert. To improve the training efficiency and performance, some improvements are also introduced into the framework. The method is validated by various experiments, including two tasks (walking as fast as possible & tracking specific velocity) and several different environments (plain, up-hill, down-hill and uneven floor). The work is conducted well and is promising in different control tasks. Both the framework and the tricks are inspiring for other works.

Experimental design

Problems in the content and explanation of the paper:
1. The description of the PPO algorithm is not detailed enough. The meaning of lambda and gamma in Table1 and their usage are not illustrated.
2. The experiments are rich for the LORM models. However, the performance of reference motion should be illustrated to show the improvement or difference between the proposed algorithm and reference motion.

Validity of the findings

Other details should also be edited:
1. Some paragraphs have indentation while others do not. For example Line 295 and Line 297; Line 273 and Line 276.
2. Please avoid capital letters in sentences. For example, Line 15 “ Learn and Outperform the Reference Motion (LORM), an RL based framework ... “ in the abstract.
3. Equation 17: f(x) = ..., however, there is no variable x or function f(x). I think it should be Criterion = ... .
4. The names of curves in Figures can be polished, for example, Fig11. And the captions can be used in figures to make the meaning of curves more clear.
5. The language should be further polished. Especially in the subsection “Symmetrization of actions and states”, I wonder whether it can be more clear? Though the description is understandable, it takes time to read and understand it.

Additional comments

In addition, I have two questions to be answered by the authors:
1. In the subsection “Symmetrization of actions and states”, why only the angles of joints are symmetrized while other observations keep unchanged?
2. The input observation of the agent contains many items which can be obtained in simulation software (base position, position of centre of mass). However, is it possible to obtain them in the real world?

Reviewer 2 ·

Basic reporting

An RL based framework for gait controlling of biped robot is proposed in this paper to overcome the complications of dynamics design and calculation. The results validated the efficiency and the advantages of the proposed method. However, there are several suggestions for the authors
1. As the proposed method is claimed to be a novel method, there should be more literature discussion in the introduction part to clarify the state-of-art of the field and thus the novelty of the paper.
2. In the result and discussion part, it is better to compare and validate the result with published works to make it more convincing.
3. There are some typo and grammar errors in the paper, please give it a proofreading for the language check.
The results are sufficient enough to validate the aim of the paper. However, more discussions are expected to emphasize the novelty and significance of the method.

Experimental design

In the result and discussion part, it is better to compare and validate the result with published works to make it more convincing.

Validity of the findings

As the proposed method is claimed to be a novel method, there should be more literature discussion in the introduction part to clarify the state-of-art of the field and thus the novelty of the paper.

---

## Round 0.2 · accepted · Accept

Based on the reviewers' reports, the revised manuscript has been improved in terms of quality and readability. Since the contribution is solid, the manuscript is acceptable for publishing in the journal.

Reviewer 1 ·

Basic reporting

The manuscript has been well-revised. Literature review is sufficient with good background provided. The results contains clear definition of all terms.

Experimental design

The experiment design is clear with sufficient details and justified. The manuscript has met the standards of the journal.

Validity of the findings

The findings have been rigorously validated with sufficient details. Conclusions are well stated.

Reviewer 2 ·

Basic reporting

all the issues I raised have been addressed

Experimental design

all the issues I raised have been addressed

Validity of the findings

all the issues I raised have been addressed

Additional comments

all the issues I raised have been addressed